# Impact of a Novel Nano-Protectant on the Viability of Probiotic Bacterium *Lactobacillus casei* K17

**DOI:** 10.3390/foods10030529

**Published:** 2021-03-04

**Authors:** Jinsong Wang, Lanming Chen

**Affiliations:** 1Laboratory of Quality & Safety Risk Assessment for Aquatic Products on Storage and Preservation (Shanghai), Ministry of Agriculture, College of Food Science and Technology, Shanghai Ocean University, 999 Hu Cheng Huan Road, Shanghai 201306, China; jswang@jcut.edu.cn; 2College of Bioengineering, Jingchu University of Technology, 33 Xiang Shan Road, Jingmen 448000, China

**Keywords:** *Lactobacillus casei*, protectants, probiotics, phytoglycogen nanoparticle, pickering emulsions

## Abstract

Probiotics are considered as desirable alternatives to antibiotics because of their beneficial effects on the safety and economy of farm animals. The protectant can ensure the viability of probiotics, which is the prerequisite of the beneficial effects. The objective of this study was to evaluate the effects of a novel nano-protectant containing trehalose, skim milk powder, phytoglycogen nanoparticles, and nano-phytoglycogen Pickering emulsions on the viability of *Lactobacillus casei* K17 under different conditions. The results indicated that the optimal concentration of the carbohydrate substrate was determined to be 10% skim milk powder (*w*/*w*) instead of trehalose. The combination of 10% skim milk powder (*w*/*w*), 1% phytoglycogen nanoparticles (*w*/*w*), and 10% Pickering emulsions (*w*/*w*) was selected as the optimal component of the protectant. Trilayer protectants with an optimal component had a more significant protective effect on the bacteria than that of the monolayer and bilayer protectants, or the control in feed storage, freeze-drying, and simulated gastrointestinal environment. A scanning electron microscope was used to monitor the morphological characteristics of the protectants for different layers on *L. casei*. In conclusion, the trilayer protectant exhibited a substantial effect on *L. casei* during storage and consumption, which could be used in the feed and functional food.

## 1. Introduction

Antibiotics play an important role in in prophylaxis and therapy as well as in the stable and sustainable development of the pastoral livestock industry [1,2]. Misuse of antibiotics as prophylactic agents is common in disease prevention, but it may lead to the development of antibiotic resistance [3,4]. After bans on the use of antibiotics in livestock were imposed in many countries (European Union banned avoparcin in 1997, bacitracin, spiramycin, tylosin and virginiamycin in 1999) [5,6,7], researchers have pursued the use of probiotics (*Staphylococcus succinus*, *Enterococcus faecium*, *Lactobacilli salivarius,* and *Lactobacilli agilis*) [8,9] as alternatives to antibiotics, because they are safe for use and have beneficial effects. Previous studies have demonstrated that probiotics increased weight gain [10], improved the diversity of intestinal microbiota [11], reduced antibiotic resistance [12] and enhanced the immune system [13] in farm animals. So, they have attracted increasing global attention. *Lactobacillus casei*, an important member of lactic acid bacteria has become one of the most used probiotics in the probiotic family [14], and China has approved *L. casei* as a feed additive in 2013 [15]. Currently, the focus is on the commercialization of probiotics and the maximization of industrial production.

Several techniques have been developed to ensure the viability and stability of probiotic products, of which drying methods are considered the most common and convenient technology for preserving and handling microorganisms [16,17]. However, the high processing temperature in the course of the manufacturing process may cause lots of undesirable loss of the probiotic strains. In modern food technology, freeze-drying has been shown to be a promising technique to enhance the storage stability of probiotics for decades [18,19]. However, processing conditions such as the low freezing temperatures or dehydration during freeze-drying may cause unwanted damage to viable probiotic strains [20,21,22]. To overcome this problem, a wide variety of cryoprotectants have been employed to increase the viability of probiotic bacteria during the freeze-drying procedure [23,24]. Skim milk powder [25], whey proteins [26], sugars [27], and many other materials have been investigated for their protective effects [28,29].

The quantities of viable and stable microorganisms are critical criteria for a successful probiotic product. According to the World Health Organization, a sufficient number of probiotic cells (10^6^–10^7^ CFU/g or mL) are a prerequisite for conferring healthy benefits to the body [30]. Thus, an adequate number of viable microorganisms serves to protect the interests of both the producer and consumer. The introduction of protectants could address this problem, effectively protect the cells, guarantee viability and survival rate, and improve application prospects. Suitable protectants provide physical support for the wrapped bacteria, reduce chemical damage, and maintain the original viability and cell characteristics during processing, storage, and subsequent consumption.

Furthermore, while most studies report solely on the effect of probiotics [31], recent studies have shown that the combination of protectants and probiotics, known as synbiotics, can improve probiotic proliferation in the intestine and help in modifying the gut community [32]. The protectants, such as prebiotics, are usually nitrogen source, carbon source and other materials (e.g., skim milk powder, sucrose, and trehalose) [33]. Prebiotics can also be used as energy sources, metabolic substrates, and micronutrients [34]. The utilization of prebiotics might maximize the effect of the products, and thus form a symbiotic combination because of their synergic effects [35].

Therefore, the objective of this study was to investigate the effect of a protectant on *L. casei* K17, a kind of *Lactobacillus* strain isolated from kimchi in a previous study [36]. First, we optimized and determined the methods of colony counting and the component of protectants. Then, to further exploit their symbiotic behavior, they were combined with *L. casei* K17 and evaluated for stability during feed storage and freeze-drying. Furthermore, gastric and intestinal simulation tests were used to study their tolerance to gastrointestinal stresses.

## 2. Materials and Methods

### 2.1. Bacterium and Culture Conditions

Before use, the *L. casei* K17 was reactivated and standardized. Then, they were inoculated into 5 mL sterilized de Man Rogosa Sharpe (MRS) media (Hangzhou Microbial Reagent Co., Ltd., Hangzhou, China), and incubated at 37 °C for 18 h as described previously [36]. After centrifugation at 3000× *g* for 10 min at 4 °C, *L. casei* K17 cells were collected, and diluted to 10^9^ CFU/mL with sterile normal saline using the colony counting method [37] (Figure 1A), and then used for further analysis. 

### 2.2. Preparation of Pickering Emulsions 

The phytoglycogen nanoparticles were isolated based on the previous study with slight modifications [38]. A 300 g of sweet corn seeds (Beijing Huanai Agricultural Development Co., Ltd., Beijing, China) were mixed with the deionized water (*w*:*w* = 1:5), then soaked at 4 °C for 24 h before being ground into powder by a multi-functional blender (Yongkang Boou Hardware Products Co., Ltd., Yongkang, China). The slurry was filtered through 100-mesh sieves and then centrifuged at 10,000× *g* for 10 min. Three volumes of ethanol were added to precipitate the soluble starch particles. After three rounds of centrifugation and decantation, the precipitate was collected and freeze-dried at −55 °C under 10 Pa for 24 h by a vacuum freeze-dryer (Shanghai Ba Yi Industrial Co., Ltd., Shanghai, China) to remove the residual ethanol. The obtained material was water-soluble phytoglycogen nanoparticles (Figure 1B). Octenyl succinic anhydride (OSA; Aladdin Reagent (Shanghai) Co., Ltd., Shanghai, China) was diluted in isopropanol (Aladdin Reagent (Shanghai) Co., Ltd., Shanghai, China) (*w*:*w* = 1:5) and added to an aqueous suspension of starch (30%, *w*/*w*) with agitation for 2 h at a level of 10%, while the pH was maintained at 8.5 using a 3 mol/L NaOH solution. After the reaction at 35 °C for 8 h, the pH value was adjusted to 6.5 using 2.5 mol/L HCl. The starch suspension was centrifuged at 3000× *g* and washed thrice with distilled water and ethanol alternately. The collected material was hydrophobic phytoglycogen nanoparticles and was preserved for further analysis.

Pickering emulsions were prepared according to the method described by Fan et al. with simple modifications [39]. The hydrophobic phytoglycogen nanoparticles were dispersed in deionized water at a concentration of 5% (*w*/*w*). Then the solution was mixed with medium-chain triglyceride (MCT; Homan Biotechnology (Shanghai) Co., Ltd., Shanghai, China) at pH 7.0 using an FJ-200 high-speed dispersion homogenizer (Shanghai Specimen Model Factory, Shanghai, China) at 12,000× *g* for 4 min to obtain 50% (*w*/*w*) phytoglycogen nanoparticles Pickering emulsions.

### 2.3. Preparation of Compound Protectants

Sterile normal saline was used as a control group in this study. Four experimental protectants were prepared as described previously and their components are listed in Table 1. The skim milk powder (BBI Life Sciences Co., Ltd., Shanghai, China) and trehalose (Sangon Biotech (Shanghai) Co., Ltd., Shanghai, China) were sterilized by heating at 65 °C for 30 min, and water-soluble phytoglycogen nanoparticles and Pickering emulsions were filtered through membrane filters (0.22 um pore size, Millipore, Bedford, MA, USA) to sterilize. The trilayer compound protectants of *L. casei* K17 were formed by regulating the pH to 7.0 and homogenizing at 12,000× *g* for 4 min. 

### 2.4. Bacterial Viability under Protectants during Feed Storage

A commercial feed purchased from Zhejiang Dongyu Biological Technology Co., Ltd., in Huzhou, Zhejiang Province, China was sterilized in a sterilization bag (121 °C, 20 min), dried at 65 °C for 2–3 h. Then, a 1.0 g of basic feed was weighed by electronic balance (EK813, Zhongshan Camry Electronic Co., Ltd., Zhongshan, China). The bacterial cell culture (0.1 mL) was mixed with different compound protectants and dripped on the feed by the micropipette (Eppendorf Research Plus 10–100 μL, Eppendorf, Hamburg, Germany). All the volume of 0.1 mL below are transferred by 10–100 μL Eppendorf pipette, all the volume of 0.2 mL by 20–200 μL Eppendorf pipette, and all the volume of 0.3–1 mL by 100–1000 μL Eppendorf pipette. After drying for 8 h on the clean bench (ACB-A, Esco Micro Pte Ltd., Singapore) at 30 °C, the Petri dishes were sealed, packed in a sterile sampling bag (Qingdao Hope Biotechnology Co., Ltd., Qingdao, China), stored at 4 °C, and used for further analyses. The bacterial concentration in the prepared feed containing different protectants was examined using the colony counting method [37]. Each 1 g feed was ground in a mortar, poured into a 15 mL test tube containing 9 mL sterile normal saline solution, shaken for 2 min, and then the middle part of the supernatant was chosen for enumeration. The bacterial solution was serially diluted, and 100 μL of the solution was added dropwise and plated on the MRS agar. The plates were incubated under anaerobic conditions for 48–72 h at 37 °C and the number of colonies was counted. The count of viable cells was expressed as log colony-forming units per gram (log CFU/g).

### 2.5. Survival of Bacteria under Protectants during Freeze-Drying

The protective effect of protectants on *L. casei* K17 under freeze-drying was studied as described by Carvalho [40]. The blank and the monolayer, bilayer and trilayer sample (4 mL) were placed in sterile vials and pre-freezed in a refrigerator (−80 °C, 3 h) before being transferred to a freeze-dryer (10 Pa, −55 °C, 24 h). Calculation of the lyophilized survival rate: viable bacteria were counted after rehydration of lyophilized bacteria powder. Lyophilized survival rate = the number of viable bacteria after lyophilization/the number of viable bacteria before lyophilization × 100%.

### 2.6. Microstructure of Lyophilized Bacterial Powder 

Freeze-dried samples was examined by a scanning electron microscope (SEM) according to the method described by Sathyabama et al. [9]. Briefly, the rehydrated bacteria (0.1 mol/L phosphate buffered saline, pH 7.0) were centrifuged (4500× *g*, 10 min) and fixed in 2.5% glutaraldehyde solution (4 °C, overnight) prior to dehydrating with 20%, 40%, 60%, 80% and 95% ethanol solutions respectively for 15 min. Afterwards, they were treated with 100% ethanol twice for 20 min before further centrifugation at 4500× *g* for 5 min. Finally, the lyophilized bacterial powder was cross-sectioned and mounted on stabs with an adhesive tape, and the morphological characteristics of the gold-coated cells were examined by SEM.

### 2.7. Survival of Bacteria under Protectants in the Simulated Gastrointestinal Environment

Simulated gastric juice (SGJ) and simulated intestinal juice (SIJ) were prepared based on the method described by Meira et al. with slight modifications [41]. A mixture of 16.4 mL HCl (0.1 mol/L), 10 g pepsin (Sangon Biotech (Shanghai) Co., Ltd., Shanghai, China) and 500 mL deionized water was shaken; subsequently, deionized water was added up to a volume of 1000 mL; and this solution was used as SGJ. Afterwards, the pH of SGJ was adjusted to 1.2 by 0.1 mol/HCl. A mixture of 10 g pancreatin (Sangon Biotech (Shanghai) Co., Ltd., Shanghai, China), 500 mL deionized water and 6.8 g potassium dihydrogen phosphate (Sangon Biotech (Shanghai) Co., Ltd., Shanghai, China) was shaken; subsequently, deionized water was added up to a volume of 1000 mL, and the solution was used as SIJ. The pH of the SIJ was adjusted to pH 6.8 using a 0.1 mol/L NaOH solution. 

Both solutions were filtered through a membrane (0.22 μm, Sangon Biotech Co., Ltd., Shanghai, China). Cell suspensions of 10^9^ CFU/mL (200 μL) were homogenized for 2 min in a vortex (MX-S SCILOGEX, San Diego, CA, USA) with 1000 μL of SJG and sterile normal saline (300 μL), before incubating separately for 1, 90, and 180 min at 37 °C. Neutralization with 1 mol/L NaOH to pH 7 was used to stop the enzymatic reactions. The SIJ (1000 μL), 10^9^ CFU/mL cell suspensions (200 μL), and sterile normal saline (300 μL) were homogenized for 2 min in a vortex and incubated at 37 °C for 1 and 240 min separately. The enumeration in both SGJ and SIJ was conducted by colony counting in MRS agar.

### 2.8. Statistical Analysis

All determinations were carried out in triplicate, and all results are presented as mean ± standard error (SE). Statistical analyses were conducted using SPSS version 22.0. The significant difference was defined at *p <* 0.05 using one-way analysis of variance and post hoc Duncan multiple range tests [42].

## 3. Results and Discussion

### 3.1. Optimization of Colony Detection on Feed

To obtain accurate experimental data and robust results, we also optimized the colony counting method for animal feed. Before eluting for colony counting, the feed, which was sprayed with 10^9^ CFU/mL *L. casei* K17 (0.1 mL), was ground into a powder. The feed was in the form of solid pellets, and grinding was performed to ensure that all the bacteria would be available in the solution for enumeration, which results in more accurate data. Meanwhile, to determine the optimal method of feed storage after spraying the prepared 10^9^ CFU/mL *L. casei* K17, we examined the survival rates in the dried and undried samples (as seen in Figure 2). The survival rates of both methods were more than 87% in comparison to the rates during the first seven days, during which the undried sample performed better than the dried one. This might be ascribed to the fact that the drying temperature does not damage the bacteria or that the water activity in the undried sample is higher than that in the dried one, where the microorganisms can survive better [43]. In the meantime, we also found a film of mold over the feed after the 7th day, which might be because the moisture in the undried sample was higher than that in the dried one. Therefore, we propose a protocol in which the bacteria could be used directly for seven days as a feed additive in an undried form or for a long period in a dried form. Although the *L. casei* strain does suffer from a small decrease in numbers during drying, the survival rate is suitable for the application.

### 3.2. Optimization of Protectant Component on Feed

In an effort to expand the *L. casei* K17 product range, skim milk powder, trehalose, phytoglycogen nanoparticles and Pickering emulsions were used to evaluate the effect of protectants on feed. The survival rates of the strains with the corresponding protectants are shown in Figure 3**.** Proteins can provide an additional protective effect by covering the cells and balancing the cell membranes during freeze-drying and storage [44]. Furthermore, skimmed milk powder contains several proteins, and is dried easily. In this study, skim milk powder was used to investigate protective effect on the strain at the concentration of 0%, 5%, 10%, and 15% for 12 days (Figure 3A). There was a significant difference in efficacy between the 10% concentration and the control in the test period, with the greatest significant difference on the 7th day. However, the difference in efficacy between 10% and 15% samples was insignificant during all the experiments. Thus, skim milk powder of 10% concentration was chosen for further analysis.

To determine the carbohydrate utilization and preferred substrates of strains, the same concentration of trehalose was used to study the protectant effect as shown in Figure 3B. Trehalose effectively increases bacterial survival during freeze-drying and storage [45] and is also known as an antioxidant that protects against oxidative stress [46]. However, the results in this study show that different concentrations of trehalose had no significant protective effect against *L. casei* K17, compared with the control. Since different strains differ in their abilities of circumstance response and substrate utilization [8], this strain might be less sensitive to trehalose. Thus, we made further efforts to employ the phytoglycogen nanoparticles combined with nanotechnology to determine the preferred carbohydrate (Figure 3C). It has been reported that phytoglycogen is a water-dispersible amylopectin analog with a more highly branched structure with a particle size ranging from 30 to 100 nm [39]. Phytoglycogen nanoparticles could provide more protection for probiotics with excellent film-forming and poor rehydration capacities [47]. According to Figure 3C, the survival rate of bacteria at 1% concentration was higher than that of the control in the test period and that at 0.5% concentration on the 7th day, but there was no significant difference in efficacy between the 1% and 1.5% concentration in the test period. 

Recently, Pickering emulsions have been used in functional food and drug delivery systems [48,49]. In this study, we used this novel technology in animal feed additives. From Figure 3D, we can see that during the first three days, there is no significant difference among these concentrations (0, 5%, 10%, and 15%). However, the survival rate at 10% concentration was significantly higher than that of the control and 5% concentration on the 7th and 12th day. When Pickering emulsions were scaled up to 15%, there were no changes in the survival rates of the bacteria during the period from 1st to 12th day. It may be possible that 10% skim milk powder and 1% phytoglycogen nanoparticles provide a suitable condition for the bacteria to survive. Therefore, no significant difference was observed in the first three days. With time, the effect of the Pickering emulsions on bacteria was more visible.

### 3.3. Protective Effect of Three Treatments on Feed and Evaluation of Bacterial Morphology and Shape Integrity by SEM after Freeze-Drying

The effect of the three types of protectants is shown in Figure 4 and further clarified by the SEM images in Figure 5. All the protectants had considerable impact on the bacteria compared with the control (Figure 4), which could also be observed in the SEM images (Figure 5B–D). The monolayer protectants exerted a remarkable impact on the strain from the 1st to 12th day, especially on the 12th day. The first layer of skim milk powder might be responsible for this phenomenon. It contains proteins that provide an additional protective layer for the cells [50] and stabilize membrane components [51] (Figure 5B). The bilayer protectants exerted a significant impact on the strain with an extremely significant effect on both the 1st and 12th days (Figure 4). The phytoglycogen nanoparticles could provide excellent film-forming properties by connecting with the milk globules and suitable carbohydrate utilization for the bacteria (Figure 5C). Trilayer protectants had an extensive protective effect on the bacteria compared to the control in all experiment durations (Figure 4). The trilayer protectant consisting of Pickering emulsions, phytoglycogen nanoparticles, and milk powder provides the suitable carbohydrates and nitrogen resources, as well as physical and chemical support. Pickering emulsions can enhance the vitality of bacteria [39]. The trilayer protectant produced a dense covering layer (Figure 5D) whereas the targets are more easily exposed under the monolayer and bilayer treatments.

### 3.4. Survival of L. casei K17 under Protectants against the Freeze-Drying Process 

For long-term preservation, freeze-drying is usually considered as a suitable technique to ensure high stability of the organism. We further examined and calculated the survival rates of *L. casei* K17 under different protectant treatments after freeze-drying (Figure 6). The results showed that different protectants exhibited significantly different protecting effects on the bacteria, and the trilayer protectants exhibited the maximum survival rate (73.2%). The developed protectant exhibits promising results as the novel cryo-protectants are beneficial for both the producer and consumer. 

### 3.5. Survival of L. casei K17 under Protectants against Simulated Gastrointestinal Conditions

The survival rate of probiotics at low pH values is an important factor that affects consumer satisfaction. Tolerance of free and encapsulated *L. casei* K17 to the simulated gastrointestinal environment was evaluated in this study. Figure 7 demonstrates the tolerance of the organisms to artificial gastric juice after being subjected to different treatments. There were no significant differences among the three treated samples and the control sample at 1 min. However, compared to the control, the monolayer, bilayer, and trilayer groups showed better results, with tolerances of 89.8%, 93.1%, and 93.3%, respectively, at 90 min. Among these three treatments, the trilayer group was the highest result and obviously higher than the monolayer group at 180 min (*p* < 0.05).

Tolerance of the bacteria to artificial intestinal juice was further evaluated. No significant difference was observed at 1 min in the control and test groups. However, the test groups were more stable compared to the control group, with survival rates of 95.9%, 97.5%, and 98% at 240 min, respectively (*p* > 0.05, Figure 8). Thus, the trilayer protectant performs better than the others in ensuring the desirable stability and viability for bacteria in the gastrointestinal environment. In previous studies, some conventional carbohydrates were not considered suitable protectants for resisting digestive juices, such as monosaccharides [52], maltodextrin [53], and inulin [54]. However, Tao, et al. [47] reported that skim milk-sodium carboxymethylcellulose and skim milk-sodium alginate offered excellent protection for probiotics with a final viability loss of 0.77 ± 0.12 and 0.80 ± 0.13 log CFU/g when exposed to the simulated gastrointestinal tract for 180 min. Meanwhile, Samedi et al. [55] reported that the microencapsulation of *Lactobacillus plantarum, Weissella paramesenteroides, Enterococcus faecalis,* and *Lactobacillus paraplantarum* with a coating of arrowroot starch (arrowroot 1.25 g, whey 0.25 g, bacteria 0.5 g, and water 2 mL) and maltodextrin (maltodextrin 1.25 g, whey 0.25 g, bacteria 0.5 g, and water 2 mL) proved to be an effective approach for delivering live bacteria at suitable levels to the intestines and served to maintain their viability in the simulated gastrointestinal tract.

## 4. Conclusions

In the present study, we examined and compared the effects of the different components of protectants for probiotics. Then, we determined the survival rate of the bacteria during feed storage and freeze-drying for different protectant components and obtained the optimal components. The gastric and intestinal simulation tests showed that the trilayer protectant is successful in providing the required effect. For the first time, this work demonstrated the use of phytoglycogen nanoparticles and Pickering emulsions combined with skim milk powder to protect bacteria. The synbiotic effect of strain and prebiotics should be considered as they can both provide energy and influence the intestinal microbiota in the gut. Although the tolerance to simulated artificial gastric and intestinal juices was evaluated using two tests and promising results were obtained, future studies must be conducted to monitor the viability and stability of bacteria in the gut using animal models. This study determined the cost-efficiency of protectants using inexpensive sources of prebiotics and met the commercialization requirements. The protectants are recommended for application in lactic acid bacteria feed production for farm animals as well as for use as a functional food for humans.

## Figures and Tables

**Figure 1 foods-10-00529-f001:**
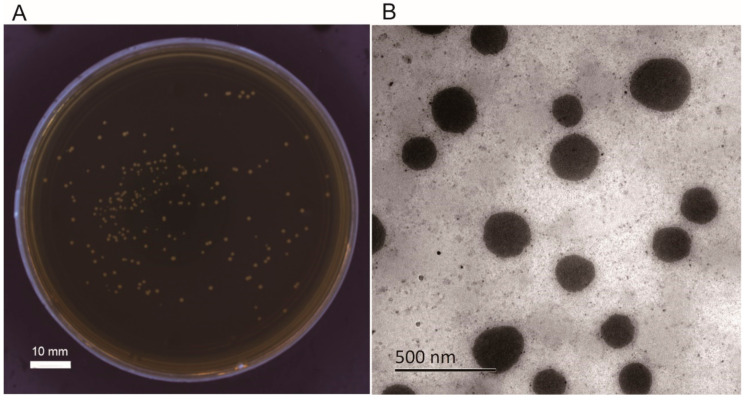
Colony counting medium and phytoglycogen nanoparticles used in this study. (**A**) Colony counting medium. (**B**) Phytoglycogen nanoparticles.

**Figure 2 foods-10-00529-f002:**
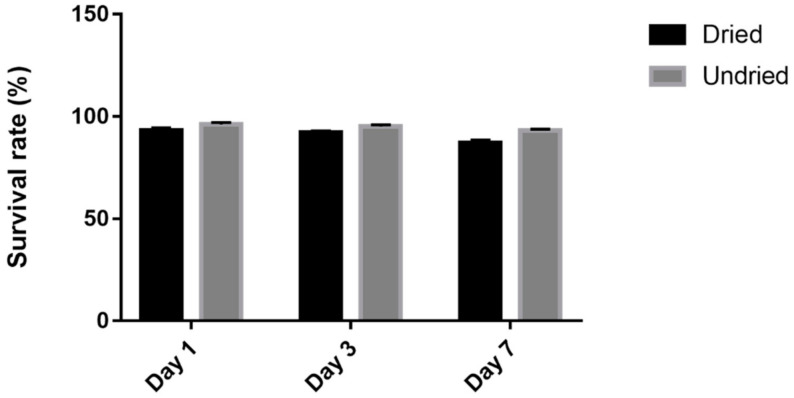
Survival rate of *L. casei* K17 on feed with dried and undried storage for 7 days. Values (mean ± SE, *n* = 3).

**Figure 3 foods-10-00529-f003:**
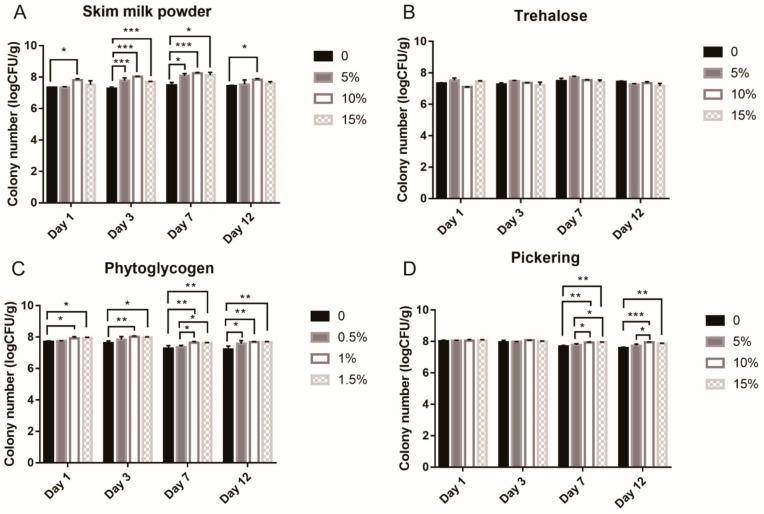
Viability of free and encapsulated *L. casei* K17 on feed during storage at 4 °C for 12 days. (**A**) Viability of *L. casei* K17 formulated with 5%, 10% and 15% (*w*/*w*) of skim milk powder. (**B**) Viability of *L. casei* K17 formulated with 5%, 10% and 15% (*w*/*w*) of trehalose. (**C**) Viability of *L. casei* K17 formulated with 0.5%, 1% and 1.5% (*w*/*w*) of water-soluble phytoglycogen nanoparticle and 10% (*w*/*w*) of skim milk powder. (**D**) Viability of *L. casei* K17 formulated with 5%, 10% and 15% (*w*/*w*) of Pickering emulsions, 10% (*w*/*w*) of skim milk powder and 1% water-soluble phytoglycogen nanoparticle. Values (mean ± SE, *n* = 3), * *p* < 0.05, ** *p* < 0.01, *** *p* < 0.001.

**Figure 4 foods-10-00529-f004:**
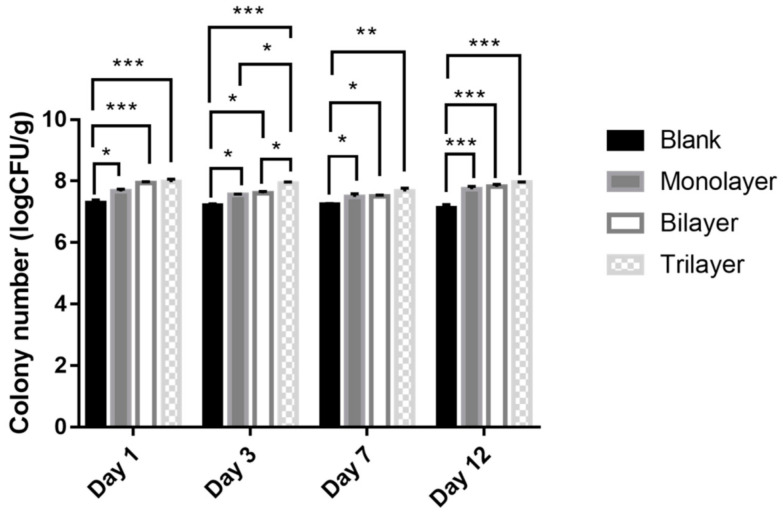
Viability of free and encapsulated *L. casei* K17 on feed during storage at 4 °C for 12 days. Values (mean ± SE, *n* = 3), * *p* < 0.05, ** *p* < 0.01, *** *p* < 0.001.

**Figure 5 foods-10-00529-f005:**
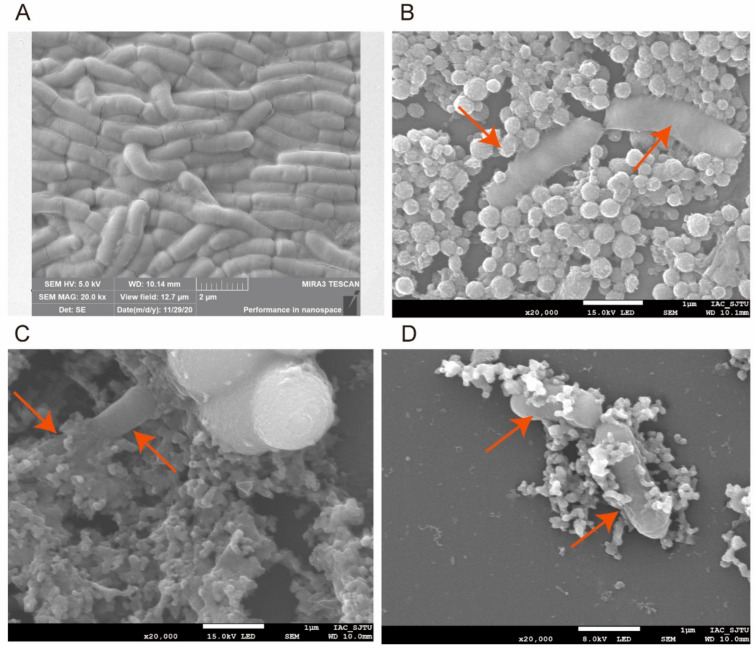
Morphology of the interaction of protectants and *L. casei* K17 cell by scanning electron micrograph. (**A**) Morphology of *L. casei* K17 cell without the addition of protectant. (**B**) Morphology of *L. casei* K17 cell with monolayer protectant. (**C**) Morphology of *L. casei* K17 cell with bilayer protectant. (**D**) Morphology of *L. casei* K17 cell with trilayer protectant.

**Figure 6 foods-10-00529-f006:**
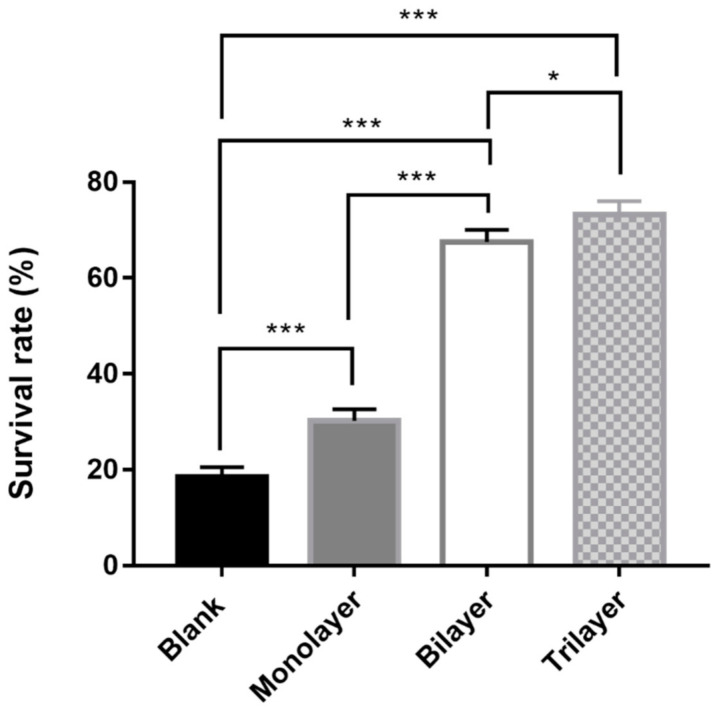
Survival rate of free and encapsulated *L. casei* K17 immediately after freeze-drying. Values (mean ± SE, *n* = 3), * *p* < 0.05, *** *p* < 0.001.

**Figure 7 foods-10-00529-f007:**
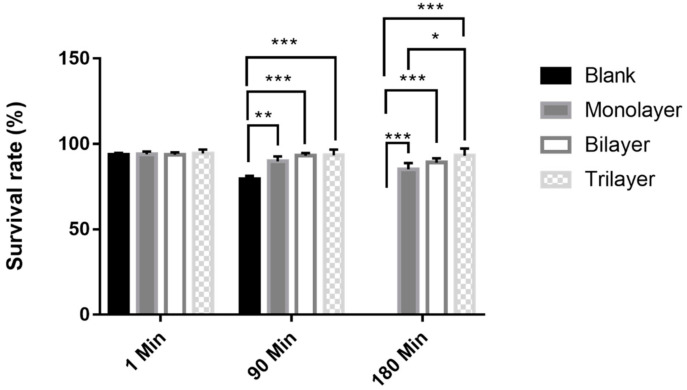
Survival rate of free and encapsulated *L. casei* K17 under simulated gastral environment for 180 min. Values (mean ± SE, *n* = 3), * *p* < 0.05, ** *p* < 0.01, *** *p* < 0.001.

**Figure 8 foods-10-00529-f008:**
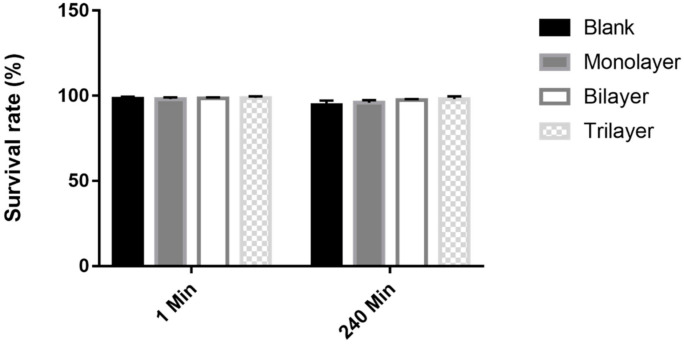
Survival rate of free and encapsulated *L. casei* K17 under simulated intestinal environment for 240 min. Values (mean ± SE, *n* = 3).

**Table 1 foods-10-00529-t001:** The different processing methods of protectants for *Lactobacillus casei* K17.

Treatment Group	Protectants Formula of *L. casei* K17
Blank	100% sterile normal saline + 10^9^ CFU/mL *L. casei* K17
Monolayer	10% sterile skim milk powder or trehalose + 90% sterile normal saline + 10^9^ CFU/mL *L. casei* K17
Bilayer	10% sterile skim milk powder + 1% sterile water-soluble phytoglycogen nanoparticles + 89% sterile normal saline + 10^9^ CFU/mL *L. casei* K17
Trilayer	10% sterile Pickering emulsions + 10% skim milk powder + 1% sterile water-soluble phytoglycogen nanoparticles + 79% sterile normal saline + 10^9^ CFU/mL *L. casei* K17

## Data Availability

The data presented in this study are available in this paper.

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
