# Peer review of "Impact of a Novel Nano-Protectant on the Viability of Probiotic Bacterium Lactobacillus casei K17"

_foods, 2021, doi:10.3390/foods10030529_

Round 1
Reviewer 1 Report
I haven't got any comments for upgrading this manuscript, as it's well-written, concise and relevant. However, there are some minor grammatical errors which should be corrected. Such as in the title - "...on THE viability of probiotic...". In the introduction, there should be "composition" instead of "component", etc. There are several errors, so please recheck grammar.
Author Response
Dear reviewer,
Thank you very much for your constructive comments. The comments are all valuable and very helpful for revising and improving our paper, as well as the important guiding significance to our researches. We have considered the comments carefully and made corrections which we hope meet with approval. The amendments are highlighted in red in the revised manuscript. Point by point responses to the reviewers’ comments are listed, please see the attachment.
Please contact me if anything else is needed. Many thanks!
Best regards,
Yours sincerely,
Jingsong Wang, PhD candidate.

Reviewer 2 Report
You report about a novel nano-protectant on viability of a certain probiotic strain. Probiotics are an effective alternative to prevent livestock diseases and thus contribute significantly to a reduced use of antibiotics. This topic is important as the use of appropriate preservatives is essential to maintaining the effectiveness of probiotics.
The study was described well structured and the results were presented as well as discussed sufficiently - nevertheless, the following points have to be revised for publication:
The method part should be presented more clearly in relation to the study plan and the sampling plan used in order to be able to understand how the experiments were carried out. Therefore, the accuracy of the weighing-in values ​​and the added volumes must be specified where necessary in order to obtain reproducible results, e.g. 0,1 mL ± 0,001 mL. The number of replicates at each of the experiments must be stated all over, both in the method section and in the presentation of the results as already done at figure 4 and 5.
Section 3.1.1 Figure 2 and description is from my point of view not necessary, because it is well known that grinding of sample material enhances significately the recovery. But it should be notet, that grinding is an essentual step for getting the correct survival rate. Furthermore, a longer shelf life after spray drying is very important for practical use of such materials - you investigated the viability after 1, 3, and 7 days. In terms of food safety, the observed mold formation within 7 days makes the application of a drying step necessary. Within this context, the comparison between dried and non-dried material is not really meaningful. In order to recommend the dried variant for longer periods, it is necessary to examine the survival rate for this possibility. The survival rate of various protective agents is then examined over a period of 1, 3, 7 and 12 days. A uniform procedure with regard to the test arrangement and length would be advantageous for comparability. At figure 7, 8, and 9 error bars should be shown for each of the results and the number of replicants should be given as already done at figure 4 and 5 in order to provide meaningful results. Significances should be given by indicating the p-values. To show the blank value as extra bar is not necessary from my point of view because this was set to 100%. Again, investigation intervals of the results shown at figure 8 and 9 should be better uniform.
Author Response

(The authors gave the same response as above.)

Round 2
Reviewer 2 Report
Dear authos,
The quality of the paper has been improved, most of the recommendations have been taken into account, especially, the idea with the graphical abstract is very useful for getting an overview on the experiment. However, there are still a few points to consider. I am referring to the revised version with track changing:
Line 304 … were sterilized by pasteurization.
Please spezify which pasteurization conditions you used.
Lines 325-327 The bacterial cell culture (0.1 mL) was mixed with different compound protectants and sprayed on the basic feed (1 g) by the micro pipette (Eppendorf, Hamburg, Germany).
What type of Eppendorf micropipette was used for transferring the 0.1 ml and how exactly was the 1 g weighed (and with which type of balance) - please state, as this is important for the accuracy of your preparations!
Lines 394-399 Cell suspensions of 109 CFU/mL (0.2 mL) were homogenized for 2 min in a vortex (MX-S SCILOGEX, Rocky, USA) with 1 mL of SJG and sterile normal saline (0.3 mL), before incubating separately for 1, 90, and 180 min at 37 °C. Neutralization with 1 mol/L NaOH to pH 7 was used to stop the enzymatic reactions. The SIJ (1 mL), 109 CFU/mL cell suspensions (0.2 mL), and sterile normal saline (0.3 mL) were homogenized for 2 min in a vortex and incubated at 37 °C for 1 and 240 min separately.
Please indicate the accuracy of the transfered volumes of 0.2 mL, 0.3 mL as well as 1 mL. In case of using again an Eppendorf pipette please indicate which type.
Line 402 All experiments were repeated three times with duplicate samples …
Please indicate what you mean „with duplicate samples“ – do you mean the pouring of two petri dishes per dilutionsstep (because you indicated later (n = 3) in the figure legend)? Please keep in mind that n has to be expressed in italic letters.
Lines 974-976 However, the test groups were significantly more stable compared to the control group, with survival rates of 95.9%, 97.5%, and 98% at 240 min, respectively (Figure 8)
Please indicate the significances between the bars in Figure 8 – it seems that there are no differences between the free and the encapsulated L.casei?
Author Response
Dear reviewer,
Thank you very much for your earnest work on our manuscript foods-1072769. Your comments are all valuable and very helpful for revising and improving our paper. We have considered the comments carefully and made corrections which we hope meet with approval. The amendments are highlighted in red in the revised manuscript. Point by point responses to the reviewers’ comments are listed, please see the attachment.
Please contact me if anything else is needed. Many thanks!
Best regards,
Yours sincerely,
Jingsong Wang, PhD candidate.
